# Disentangling Reasoning Tokens and Boilerplate Tokens For Language Model Fine-tuning

## Abstract

When using agent-task datasets to enhance agent capabilities for Large Language Models (LLMs), current methodologies often treat all tokens within a sample equally. However, we argue that tokens serving different roles—specifically, reasoning tokens versus boilerplate tokens (*e.g.*, those governing output format)—differ significantly in importance and learning complexity, necessitating their disentanglement and distinct treatment. To address this, we propose a novel *Shuffle-Aware Discriminator* (SHAD) for adaptive token discrimination. SHAD classifies tokens by exploiting predictability differences observed after shuffling input-output combinations across samples: boilerplate tokens, due to their repetitive nature among samples, maintain predictability, whereas reasoning tokens do not. Using SHAD, we propose the *Reasoning-highlighted Fine-Tuning* (RFT) method, which adaptively emphasizes reasoning tokens during fine-tuning, yielding notable performance gains over common Supervised Fine-Tuning (SFT).

## 1 Introduction

Recently, there has been a surge of enthusiasm in researching Agents based on Large Language Models (LLMs) (Weng, 2023; Wang et al., 2024), with the aim of achieving human-level artificial intelligence or beyond. Despite LLMs showcasing remarkable capabilities in various areas, they have not inherently demonstrated strong agent capabilities, such as multi-step reasoning (Wei et al., 2022; Yao et al., 2023; Qiao et al., 2024) and tool use (Qin et al., 2024; Schick et al., 2023; Liu et al., 2024; Patil et al., 2023). This shortfall has directed significant attention toward incorporating datasets tailored for agent tasks to enhance the agent capabilities of LLMs (Chen et al., 2023; Zeng et al., 2023; Chen et al., 2024b). These datasets offer **structured** examples of standard reasoning chains for solving agent tasks (Chen et al., 2024b; Qin et al., 2024), enabling LLMs to learn from them and thereby enhance their agent capabilities.

When leveraging these datasets to bolster LLMs' agent capabilities, existing research often treats all tokens within a sample equally (Chen et al., 2023; Zeng et al., 2023; Chen et al., 2024c; Qin et al., 2024). However, we argue that these tokens could differ substantially in learning difficulty and importance. Given the standardized structure of the data, tokens within a sample can be divided into two categories as depicted in Figure 1: 1) boilerplate tokens, which include format tokens that constrain the output structure, and template-connecting tokens that serve as standard transitional phrases for reasoning, such as "Based on the user's request... By doing so... This way..."; and 2) reasoning tokens, which provide sample-specific reasoning information crucial for task solving. Boilerplate tokens are distinctly less critical for task solving compared to reasoning tokens and are easier to learn due to their repetitive nature across many samples.

It is crucial to distinguish between the reasoning and boilerplate components and handle them separately. Failure to do so may result in undesired effects, such as overfitting to the boilerplate components, as depicted in Figure 1, ultimately leading to inadequate agent capabilities. While manually crafting regular expressions to filter out boilerplate tokens appears to be a feasible solution, it can be highly inefficient when dealing with data of diverse formats. Additionally, creating regular expressions for template-connecting tokens of transitional phrases poses challenges due to their potential variability in language. Therefore, an automated and adaptive approach for segregating these components is highly desirable.

Thought: Based on the user's request to find the most popular genre in the Media-Group tool, I should call the "list_genres_for_media_group" function to retrieve a list of genres. By doing so, I can analyze the genres and determine which one is currently trending based on popularity. This way, I will be able to provide the user with the information they are looking for regarding the most popular genre in the Media-Group tool.
Action: list_genres_for_media_group
Action Input: {}

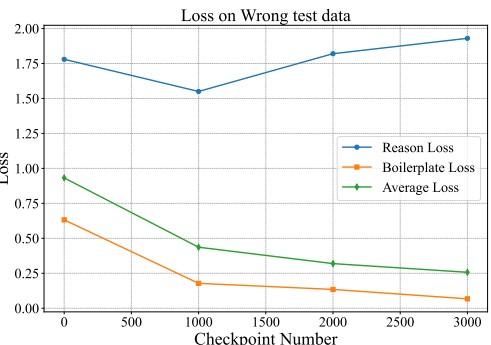

Figure 1: **Left**: Examples of reasoning tokens (green) and boilerplate tokens (yellow and blue). Boilerplate tokens can be further categorized into format tokens (yellow) and template-connecting tokens (blue). **Right**: Loss changes for different types of tokens in the sampled test data that the model fails to answer for the regular SFT training.

This study introduces a novel *SHuffle-Aware Discriminator (SHAD)* to achieve automated and adaptive token distinction. Considering boilerplate tokens are usually consistent across samples, they can be treated as sample-independent. Consequently, shuffling the correspondence between input and output across data samples does not alter the predictability of boilerplate tokens. However, such shuffling introduces noise that complicates the prediction of reasoning tokens, by causing mismatches between the tokens and the input queries[1]. SHAD is developed based on this principle. Specifically, it fine-tunes an LLM model using a small portion of shuffled data and then compares the token-level loss between the tuned and original models to classify tokens for the target data. A token is classified as boilerplate[2] if the loss on the tuned model decreases; otherwise, it is classified as a reasoning token.

Based on SHAD, we have developed a new Reasoning-highlighted Fine-Tuning (RFT) approach, which adaptively assigns greater weights to challenging reasoning tokens to emphasize the learning of reasoning. This approach demonstrates superior performance compared to existing supervised fine-tuning methods across several common agent benchmarks. Further analysis reveals that our method could effectively identify reasoning tokens and strengthen the learning of these tokens, ultimately enhancing the learning of agent capabilities for LLMs.

The main contributions of this work are summarized as follows:

- We emphasize the differences in learning difficulty and importance between reasoning and boilerplate tokens for agent learning, highlighting the critical importance of effectively distinguishing between them.
- We introduce SHAD, a novel method that automatically discriminates between reasoning and boilerplate tokens based on their predictability differences observed after shuffling input-output combinations.
- We have developed a new fine-tuning method RFT rooted in SHAD, distinctly improving the effectiveness of learning agent capabilities for LLMs.

## 2 RELATED WORK

### 2.1 TOKEN DIFFERENTIATION

Typically, when tuning LLMs, sequence-level loss is optimized, treating all involved tokens equally. However, recent studies across various domains have increasingly recognized that tokens play different roles. For instance, Lin et al. (2024) suggest that not all tokens are necessary during pretraining,

---

[1]We will later provide practical examples in Section 3.1 to illustrate how shuffling can cause the reasoning parts of a response to mismatch with the corresponding queries.

[2]These tokens would be further categorized into formatting tokens and template connecting phrases based on their losses, if need.

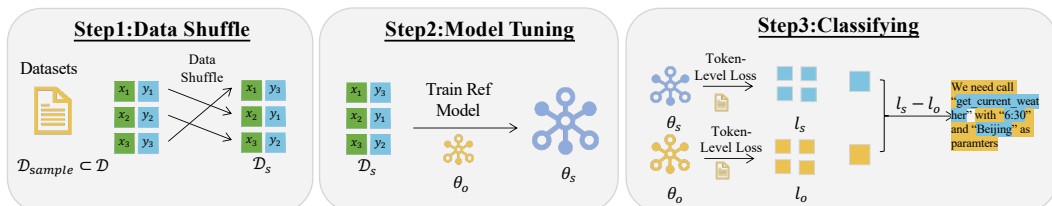

Figure 2: Illustration of the proposed SHAD method, which classifies tokens through three steps. In step 1, a small subset of the data is sampled, and the output of the sampled data is shuffled. In step 2, the LLM is tuned using the shuffled data. In step 3, tokens are classified by comparing the prediction losses between the tuned and original models.

especially in domain-specific contexts, and propose leveraging a reference model trained on high-quality data to distinguish between token importance. Similarly, Yang et al.; Rafailov et al. (2024) recognize token differences in preference learning for LLMs, and accordingly introduce token-level rewards to better align models with human preferences. Among existing works, Agent-Flan (Chen et al., 2024b) is the most relevant to ours, sharing a similar motivation to account for token differences in agent tuning. However, it only considers "format tokens" as boilerplate tokens, overlooking template-connecting tokens, which are more challenging to disentangle from reasoning tokens. Additionally, it does not emphasize the importance of distinguishing (or classifying) these tokens, resulting in a fundamental difference in both the problems addressed and the solutions proposed. We focus on automatically disentangling reasoning tokens from boilerplate tokens, whereas Agent-Flan prioritizes converting agent data into a standard conversational format.

## 2.2 ENHANCING AGENT CAPABILITY FOR LLMS

To tackle complex real-world problems, it is essential to enhance LLMs' agent capabilities, such as the ability of external tool use and multi-step reasoning (Shen et al., 2023; Nakano et al., 2021; Yao et al., 2022). Prior works (Yao et al., 2023; Shinn et al., 2023; Pan et al., 2024; Zhao et al., 2023) have focused on developing frameworks that prompt LLMs to integrate tools better and engage in deeper reasoning before taking action. Subsequent works have further constructed diverse and well-structured agent-task benchmark datasets, *e.g.,* Toolllama (Qin et al., 2024), Toolalpaca (Tang et al., 2023), and APIGen (Liu et al., 2024), considering these specific datasets for further tuning of LLMs to more directly and effectively enhance their agent abilities. Recently, Chen et al. (2024b) proposed Agent-Flan, a dataset rewrite method to enable LLMs better to learn reasoning and tool use at a step level. Although these methods train LLMs on agent datasets and achieve promising results, they often struggle with overfitting and generalization issues (Chen et al., 2024b). Our RFT with SHAD can better utilize these datasets to learn reasoning, achieving superior performance on agent tasks while maintaining good generalization ability on out-of-distribution benchmarks.

## 3 METHODOLOGY

In this section, we first introduce the SHuffle-Aware Discriminator (SHAD), which is proposed to adaptively distinguish between reasoning and boilerplate tokens. We then discuss how to develop our Reasoning-highlighted Fine-Tuning (RFT) based on the discrimination results.

### 3.1 SHAD: ADAPTIVE TOKEN DISCRIMINATOR

To develop SHAD, our foundational idea is that boilerplate tokens, which template outputs, should be interchangeable across many samples, whereas reasoning tokens are specific to individual samples and cannot be swapped. Consequently, shuffling the combination of inputs and outputs across samples does not alter the predictability of boilerplate tokens, unlike reasoning tokens. Leveraging this principle, we could achieve automated and adaptive token discrimination through the tree steps (as show in Figure 2):

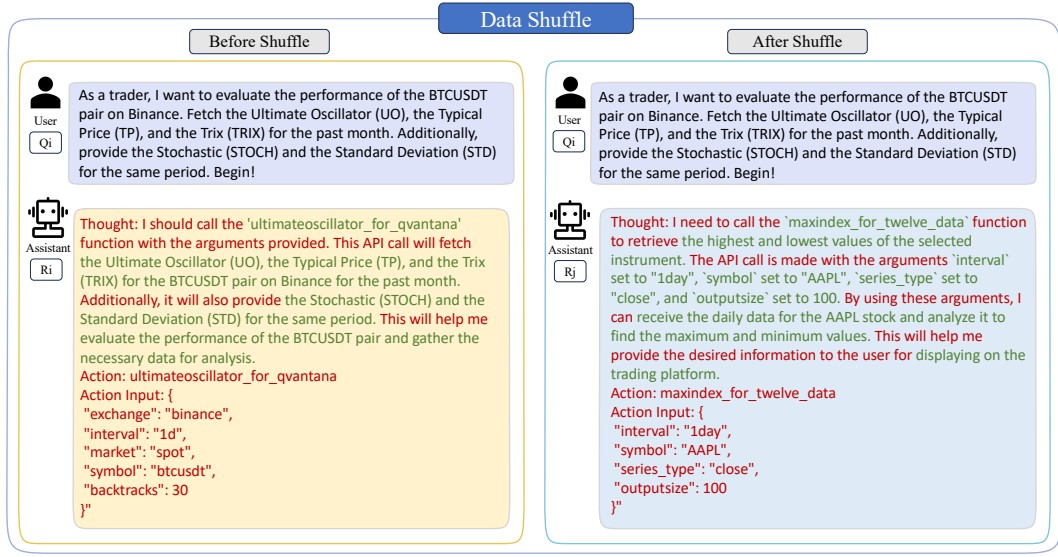

Figure 3: Example of shuffled data. After shuffling, the assistant's responses no longer correspond to the original queries. However, some tokens (boilerplate tokens, red) remain semantically similar to the original response and are therefore predictable. In contrast, reasoning tokens (green) no longer align with the query, resulting in noise. Note that 'Action' and 'Action Input' are directly copied from 'Thought' and could be considered as non-reasoning.

1. **Data Shuffle**: Select a small ratio of the data and shuffle the combinations of inputs and outputs among the sampled items.
2. **Model Tuning**: Fine-tune an LLM model using the shuffled data.
3. **Classifying**: Classify tokens based on the loss change between the tuned and original models for the target data. Compared to the original model, if a token's loss decreases, it is likely a boilerplate token; otherwise, a reasoning token.

Next, we elaborate on these three steps:

•**Data Shuffle.** This is the core step of our method, creating distinct predictability for the reasoning tokens and boilerplate tokens. The shuffle is performed by randomly reassigning the input-output combinations between samples. When implementing, we just select a small ratio (1%) of the target dataset and shuffle it for use in the subsequent model tuning step, to avoid large tuning costs and overfitting on the whole dataset.

Let $(x^i, y^i)$ denote the $i$-th sample for the sampled dataset, with $x^i$ as the input and $y^i$ as the output. Denote all the inputs of all samples as $X = [x^1, \ldots, x^N]$, and the corresponding outputs as $Y = [y^1, \ldots, y^N]$, where $N$ denotes the size of sampled dataset. We shuffle $Y$, and then re-combine the inputs in $X$ and outputs in the shuffled $Y$ to construct the shuffled dataset $\mathcal{D}_s$. This means, for the $i$-th original sample $(x^i, y^i)$, its input $x^i$ may be combined with the $j$-th sample's output $y^j$ to form a new sample $(x^i, y^j)$, while its output $y^i$ may be combined with the $k$-th sample's input $x^k$ to form a new sample $(x^k, y^i)$. With this operation, the mapping relationship between the inputs and outputs becomes noise for reasoning tokens, making them unpredictable. As for the boilerplate tokens, since they are shared across samples, their predictability remains intact. Figure 3 provides an example to illustrate this.

•**Model Tuning.** After obtaining the shuffle data, we leverage them to fine-tune an LLM model. The model tuning is performed according to the classic causal language modeling. Formally,

$$\theta_s = argmin \sum_{(x', y') \in \mathcal{D}_s} l(x', y'; \theta), \qquad (1)$$

where $\theta$ denotes the learnable model parameters, and $l(x'; y'; \theta)$ denotes the loss for a shuffled sample $(x', y') \in \mathcal{D}_s$, and $\theta_s$ denotes the optimized $\theta$. As the output is shuffled for the input, the tuned model is only expected to learn to predict boilerplate tokens effectively.

•**Classifying.** After tuning the model with shuffled data, we evaluate the role of each token in a target sample by comparing the prediction loss between the tuned and original models. Given that the tuned model should primarily learn boilerplate tokens, we classify a token as 'boilerplate' if its prediction loss decreases in the tuned model relative to the original; otherwise, we classify it as a 'reasoning' token.

Given a sample $(x, y)$ in the target dataset, we focus on classifying the tokens in the output part. Formally, for the $k$-th token $y_k$ in the output, we first compute the prediction loss difference (denoted as $LD(y_k)$) between the tuned and original models as follows:

$$LD(y_k) = l_s(y_k) - l_o(y_k), \tag{2}$$

where $l_s(y_k)$ and $l_o(y_k)$ represent the loss calculated on the tuned model and the original model, respectively, given by:

$$l_s(y_k) = -\log(P(y_k|x, y_{<k}; \theta_s)), \quad l_o(y_k) = -\log(P(y_k|x, y_{<k}; \theta_o)). \tag{3}$$

Here, $P(y_k|x, y_{<k}; \theta_s)$ and $P(y_k|x, y_{<k}; \theta_o)$ denote the predicted probabilities of the token $y_k$ from the tuned model (parameterized by $\theta_s$) and the original model (parameterized by $\theta_o$), respectively.

Based on the calculated loss difference $LD(y_k)$, the token is classified as follows:

$$Classifier(y_k) = \begin{cases} boilerplate & \text{if } LD(y_k) \leq 0 \\ reasoning & \text{else} \end{cases} \tag{4}$$

Note that our token classification can be conducted offline with a single forward pass of LLM computation for each sample, without affecting the efficiency of the subsequent agent tuning process.

### 3.2 REASONING-HIGHLIGHTED FINE-TUNING

Agent-tuning data often follows fixed formats and similar reasoning trajectories, making the corresponding tokens (boilerplate tokens) easily learned by the model. If all tokens in a sample are treated equally, the model risks overfitting to these boilerplate tokens, which can hinder its ability to learn to reason effectively. To address this issue, we propose focusing more on reasoning tokens identified by our SHAD method, assigning them higher weights during fine-tuning to enhance reasoning capabilities for agent task solving.

Instead of manually assigning fixed weights to the two types of tokens, we utilize an adaptive weight assignment to align the dynamic learning process better. Specifically, we compare the total losses of the reasoning and boilerplate parts, applying the softmax function to assign higher weights to the part with the greater loss. Notably, since the reasoning part typically exhibits a higher loss (see Figure 5), our method naturally assigns greater weights to emphasize reasoning learning. Furthermore, when the loss difference between the two parts diminishes, our method can adaptively adjust the weights to promote a more balanced learning process for the two parts. Given the nature of highlighting reasoning, we name our method Reasoning-highlighted Fine-Tuning (RFT).

Formally, let $\mathcal{L}_b$ and $\mathcal{L}_r$ represent the total loss for the boilerplate and reasoning tokens, respectively. The re-weighted loss of our RFT, denoted as $\mathcal{L}_{RFT}$, can be formulated as follows:

$$\mathcal{L}_{RFT} = \omega_b \mathcal{L}_b + \omega_r \mathcal{L}_r, \tag{5}$$

where

$$\omega_b = \frac{\exp(\mathcal{L}_b/\tau)}{\exp(\mathcal{L}_b/\tau) + \exp(\mathcal{L}_r/\tau)}, \quad \omega_r = \frac{\exp(\mathcal{L}_r/\tau)}{\exp(\mathcal{L}_b/\tau) + \exp(\mathcal{L}_r/\tau)}. \tag{6}$$

Here, $\tau$ is the temperature coefficient of the softmax function. A smaller $\tau$ results in greater weight being assigned to the component with the higher loss.

Table 1: Performance comparison between baselines, SHAD+RFT, and its variants. Accuracy is reported for BFCL, Nexus, and T-eval, while pass rate, assessed by GPT-4, is used for StableTool-Bench. 'AVG' represents the average performance across all evaluation datasets. The best results among baselines and SHAD+RFT are highlighted in bold, and the second-best are underlined.

| Model | Method | Held-In | | Held-Out | | AVG |
|-------|--------|---------|------|--------|-------|-----|
| | | StableToolbench | BFCL | T-eval | Nexus | |
| LLaMA3-8B | SFT | 43.1 | 85.9 | 67.0 | 14.0 | 52.5 |
| | Regex | 36.2 | 81.0 | 54.3 | 6.45 | 44.5 |
| | Rho-1 | 24.5 | 82.9 | 68.4 | 19.0 | 48.7 |
| | RewardFT | 44.4 | **89.3** | 66.3 | 8.0 | 52.0 |
| | **SHAD+RFT** | **50.1** | 87.6 | **71.8** | **27.8** | **59.3** |
| | *SHAD+α-FT* | 47.0 | 87.2 | 68.8 | 28.7 | 57.9 |
| | *Regex+RFT* | 41.2 | 83.81 | 61.1 | 12.4 | 49.6 |
| LLaMA3.1-8B | SFT | 48.5 | 89.3 | 64.2 | 19.5 | 55.4 |
| | Regex | 42.3 | 82.1 | 58.6 | 14.3 | 49.3 |
| | Rho-1 | 30.6 | 84.6 | 67.0 | 26.0 | 52.0 |
| | RewardFT | 48.2 | 88.2 | 66.4 | 19.1 | 55.5 |
| | **SHAD+RFT** | **50.4** | **89.4** | **68.3** | **32.0** | **60.0** |
| | *SHAD+α-FT* | 49.2 | 88.2 | 63.8 | 28.9 | 57.5 |
| | *Regex+RFT* | 46.7 | 80.31 | 57.6 | 16.2 | 50.2 |

# 4 EXPERIMENTS

We now present experiments to evaluate the effectiveness of our method in enhancing LLMs' agent capabilities, particularly in multi-step planning and tool usage, for solving complex real-world problems. We begin by detailing the experimental setup, followed by the analyses of the results.

## 4.1 EXPERIMENT SETUP

**Training Data.** We use LLaMA3.1-8B and LLaMA3-8B as the backbone models, fine-tuning them to solve agent tasks. The training dataset is constructed from two commonly used multi-step planning and tool-use benchmarks, ToolBench (Qin et al., 2024) and APIGen (Liu et al., 2024), supplemented with general data from ShareGPT (noa, 2024b). The general data is used to preserve general capabilities like instruction-following, as demonstrated in previous work (Zeng et al., 2023). Tool-Bench and APIGen provide a variety of examples for solving complex real-world user queries across different environments, all organized in a standard agent-specific format: "Thought-Action-Action Input" in JSON.

**Evaluation Setting.** To comprehensively evaluate the proposed method, we consider two evaluation settings—held-in task evaluation and held-out task evaluation, following prior work (Zeng et al., 2023). The held-in evaluation focuses on measuring performance on tasks similar to those used during training, while the held-out evaluation assesses the model's generalization to novel tasks. For the held-in setting, we use the StableToolBench (Guo et al., 2024) and BFCL (noa) benchmarks. These datasets align with our agent tuning datasets: StableToolBench shares the same source as Tool-Bench, while BFCL serves as the leave-out evaluation data for APIGen. For the held-out setting, we use two additional benchmarks: 1) T-eval (Chen et al., 2024a), a comprehensive step-level reasoning benchmark, and 2) Nexus (team, 2023), a complex single-step nested tool-use benchmark. Both benchmarks provide a diverse set of tools for LLMs to choose from, with tasks in StableToolBench and T-eval often requiring multiple steps to complete. In accordance with the evaluation metrics established in the original benchmarks, accuracy is employed for BFCL[3], T-eval, and Nexus, while StableToolBench[4] is evaluated using the pass rate assessed by GPT-4.

**Compared Methods.** To evaluate our RFT method developed on SHAD (denoted as SHAD+RFT), we compare it against the following baselines: 1) **SFT**, standard supervised fine-tuning; 2) **Regex**,

---

[3]Accuracy is reported by abstract syntax tree evaluation for BFCL.

[4]We only select three most difficult subsets of StableToolbench — I2-Category, I3-Instruction, and I1-Tool.

which uses regular expressions to distinguish formatting tokens from other tokens and re-weights their losses with constant values; 2) **Rho-1** (Lin et al., 2024), which leverages a reference model trained on high-quality data to identify noise tokens and then mask them during fine-tuning; and 3) **Reward-based Fine-Tuning (RewardFT)** (Yang et al.; Rafailov et al., 2024), which assigns token-level reward scores for tuning using a DPO-based reward model. It is important to note that Rho-1 and RewardT were not originally designed for agent tuning tasks; however, we have extended them for this purpose, with implementation details provided in Appendix B.

In addition to the above baselines, we also compare our method with two of its variants to assess its core design components: 1) **SHAD+$\alpha$-FT**, which retains the SHAD component but assigns a fixed weight $\alpha$ to reasoning tokens to emphasize them; and 2) **Regex+RFT**, which preserves the RFT weighting mechanism but uses regular expressions for token distinction. The implementation details of $\alpha$-FT are also provided in Appendix B.

## 4.2 MAIN RESULTS

Table 1 summarizes the performance of all compared methods, where we could draw two main conclusions:

**SHAD+RFT Performs Strongly**. Our method, SHAD+RFT, outperforms all baselines on all held-in and held-out evaluation datasets, except for the held-in evaluation BFCL with LLaMA3-8B. This highlights the advantage of emphasizing reasoning components in solving complex real-world problems and demonstrates the effectiveness of our method in identifying and highlighting these parts. Notably, while Rho-1 and RewardFT also differentiate between tokens during learning, they are not specifically designed for agent tuning to discover and emphasize reasoning tokens, resulting in comparatively lower performance. Specifically, Rho-1 targets identifying noise tokens for masking during tuning but fails to distinguish between normal boilerplate and reasoning tokens. RewardFT leverages token rewards from a DPO-based reward model aligned with human preferences to differentiate tokens, yet it is also not designed to identify reasoning tokens essential for agent-specific capabilities.

**Both SHAD and RFT Are Crucial**. When comparing SHAD+RFT with its variants, Regex+RFT and SHAD+$\alpha$-FT, the original SHAD+RFT consistently demonstrates superior performance. We explained the results as follows:

- **Adaptive weighting in RFT is crucial**. Comparing SHAD+RFT with its variant SHAD+$\alpha$-FT, SHAD+RFT consistently outperforms, demonstrating the superiority of RFT's adaptive mechanism over the fixed weighting approach of $\alpha$-FT. This advantage stems from adaptive weighting's ability to better align with the dynamic learning process, adaptively adjusting weights for reasoning and boilerplate token components, thereby preventing over-learning or under-learning of either part.

- **The importance of SHAD for token differentiation.** Replacing SHAD with Regex in SHAD+RFT leads to a significant drop in model performance. This highlights that the effectiveness of reasoning-highlighted fine-tuning depends on accurate token differentiation. The results also demonstrate SHAD's superior ability to disentangle boilerplate tokens from reasoning tokens. In contrast, Regex relies solely on regular expressions to identify formatting tokens, failing to fully distinguish between template-connecting tokens (one part of boilerplate tokens) and reasoning tokens.

These indicates that replacing either SHAD or RFT diminishes the method's effectiveness, affirming the importance of both components.

## 5 ANALYSIS ON SHAD AND RFT

In this section, we first present a case study on the effectiveness of SHAD in distinguishing tokens, followed by a comprehensive analysis of how RFT functions.

**Case study of tokens classified by SHAD.** To further validate SHAD's ability to identify reasoning tokens, we conducted a series of case studies, with one example of classification result shown in

| Examples of Tokens Classified by SHAD |
| --- |
| Thought: I should call the API "smart_phones_for_amazon_api_v2" with empty arguments to fetch the top-rated smartphone options from Amazon. This API specifically caters to the task of finding top-rated smartphones, so it is the appropriate choice. By calling this API, I can retrieve the necessary information to suggest the user some of the best-rated smartphones available on Amazon. 
 Action: smart_phones_for_amazon_api_v2 
 Action Input: {} |
| { 
    "tool_calls": [ 
        {"name": "getgamelevel", "arguments": {"level": 5, "output": "json"}} 
    ] 
} |

Figure 4: Case study of tokens classified by SHAD. The blue regions represent reasoning tokens, identified by an increase in loss on the model tuned with shuffled data compared to the original model. In contrast, the brown regions indicate boilerplate tokens, characterized by a decrease in loss on the tuned model.

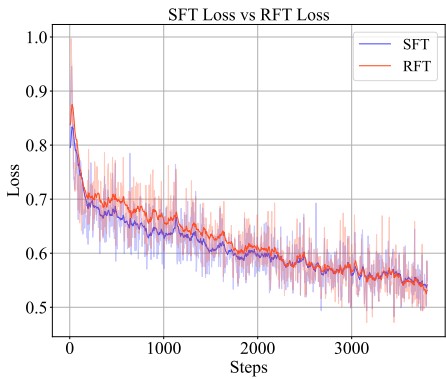
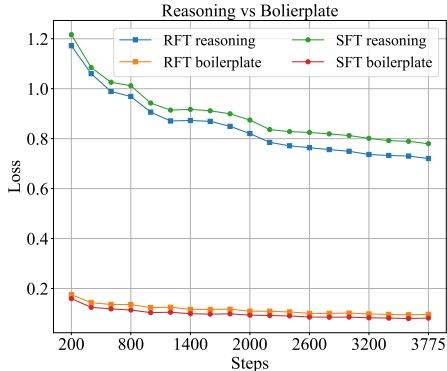

Figure 5: Training loss for SFT and our RFT (based on SHAD). **Left**: Overall training loss; **Right**: Training loss for reasoning token part and boilerplate token part.

Figure 4 (additional examples are provided in Appendix C). As shown in the figure, SHAD successfully classifies most query-dependent information—information related to 'smart-phones'—as reasoning tokens, while formatting tokens (*e.g.*, the JSON-style attribute names 'Thought' and 'Action') and common template-connecting tokens like 'I should call' and 'this API' are classified as boilerplate tokens. This outcome aligns with human understanding of reasoning tokens, verifying the effectiveness of our method again. Interestingly, SHAD does not classify the entire function name 'smart_phones_for_amazon_api_v2' as reasoning but only the 'smart_phones' portion. We think this is may because the 'amazon_api_v2' part is common across many function names. Additionally, in the 'Action: smart_phones_for_amazon_api_v2' field, the function name is not classified as reasoning, possibly because it is derived from the thought process rather than reflecting core reasoning.

**RFT Enhancing Reasoning Token Learning**. Blindly treating reasoning and boilerplate tokens equally, as done in SFT, can lead to overfitting on boilerplate tokens while insufficiently learning for reasoning tokens. To address this issue, we propose RFT based on SHAD to prioritize the learning of reasoning tokens during fine-tuning. To further verify the effectiveness of RFT, we compare the training loss between SFT and RFT. The results are summarized in Figure 5. The findings indicate that RFT significantly reduces the loss for reasoning tokens while maintaining a comparable loss for

---

Example on Held-Out Benchmark Nexus

**User Question:**
Can you please find a list of common vulnerabilities and exposures (CVE) that have a CVSSv2 severity of 'high' and then filter this list for those that contain a description in Spanish?

---

**Response (SFT):**
Function call:
filterCVEsBySeverity(cveList, severityLevel='HIGH')
filterCVEByLanguage(cveList, language='es')

Explanation:
The user wants to find a list of CVEs with a CVSSv2 severity of 'high' and then filter this list for those that contain a description in Spanish. To accomplish this, we need to call the 'filterCVEsBySeverity' function with the argument 'severityLevel' set to 'HIGH'. This will filter the CVE list for CVEs with a CVSSv2 severity of 'high'. Then, we need to call the 'filterCVEByLanguage' function with the argument 'language' set to 'es'. This will filter the CVE list for those CVEs that contain a description in Spanish. By combining these two function calls, we can achieve the desired result.

- - - - - - - - - - - - - - - - - - - - - - - - - - - - - - - - - - - - - - - - - - - - - - - -

**Response (SHAD+RFT):**
filterCVEByLanguage(searchCVE(cvssV2Severity='high') , 'es')

---

Figure 6: **Comparison example on Held-Out Benchmark Nexus.** In the response generated by the naïve SFT model, we observe overfitting, with formatted tokens and template-connect tokens being erroneously generated. Additionally, the naïve SFT model exhibits hallucination, leading to reasoning errors. In contrast, our SHAD+RFT method successfully follows the Held-Out instructions and provides accurate reasoning, we explicitly mark the different reasoning part in red.

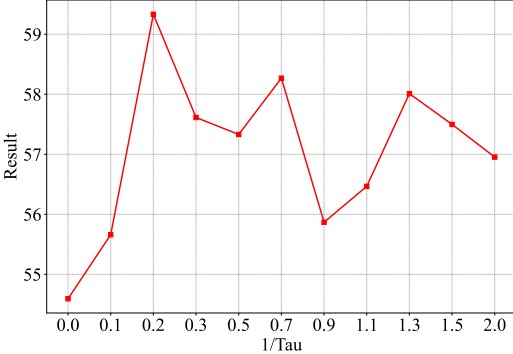

Figure 7: The performance of our SHAD+RFT method as the temperature coefficient $\tau$ varies. The performance averaged over all evaluation datasets is reported, with LLaMA3-8B as the backbone. Notably, $1/\tau = 0$ means assigning equal weights to the reasoning and boilerplate parts, *i.e.*, deactivating our re-weighting mechanism.

boilerplate tokens compared to SFT, confirming that RFT effectively enhances the learning for reasoning tokens. The empirical results on the held-out Benchmark Nexus, presented in Figure 6 (with additional examples provided in Appendix D), demonstrate that our method effectively mitigates hallucination and overfitting in boilerplate tokens, improving the model's reasoning capabilities.

**The Effect of Hyper-parameter $\tau$.** The temperature coefficient $\tau$ in Equation 6 plays a crucial role in controlling the strength of our re-weighting mechanism in RFT, so we next investigate its impact. Specifically, we vary $1/\tau$ within the range of [0, 2] and analyze the corresponding performance of SHAD+RFT (averaged over all evaluation datasets). The results are illustrated in Figure 7. From the figure, we observe that the performance of our method initially increases and then roughly decreases as $1/\tau$ increases, *i.e.*, as gradually enhancing our re-weighting mechanism. This indicates the importance of carefully selecting the optimal $\tau$. Fortunately, across a wide range, SHAD+RFT could consistently outperform regular SFT and surpass most baselines (*c.f.*, Table 1).

## 6 LIMITATION

We identify several limitations of our method in both token differentiation and re-weighting during training. First, the effectiveness of our approach depends on boilerplate tokens remaining consistent across different samples. When this consistency is lacking, such as in cases where the diversity of boilerplate tokens is high, our method may fail. Second, our distinction between reasoning and boilerplate tokens relies on rigid, manually defined thresholds for loss differences, which may need refinement. Third, our weighting strategy is currently applied only at the group level, and future optimization may be required at the token level.

## 7 CONCLUSION

In this paper, we highlight the importance of distinguishing between reasoning and boilerplate tokens and introduce a SHuffle-Aware Discriminator (SHAD) to automatically achieve this. Building on SHAD, we further developed a new Reasoning-Highlighted Fine-Tuning (RFT) method to enhance reasoning learning during LLM fine-tuning, thereby improving agent capabilities. Extensive results demonstrate that our method significantly enhances LLMs' ability to solve complex real-world problems. In the future, we plan to extend our approach to the entire SFT domain, and plan to develop more refined mechanisms, such as token-level re-weighting, to better leverage our token differentiation results.

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

## A  DETAIL INFORMATION OF TRAINING DATASETS

We provide more details of our training datasets. To enable the multi-step reasoning ability of LLM, we choose ToolBench Qin et al. (2024) and APIGen Liu et al. (2024) as our basic datasets. Following the practice in AgentTuning Zeng et al. (2023) and AgentFlan Chen et al. (2024b), we also mix ShareGPT noa (2024b) and basic datasets for training. We filter the obviously low-quality data that does not follow the request format and sample 10% percent of data from APIGen for data balance. All methods use the same dataset and do not apply token differentiation to general data.

## B  IMPLEMENTATION DETAILS

### B.1  IMPLEMENTATION DETAILS OF RHO-1

For the Rho-1 baseline, we train the reference model in self-reference setting Lin et al. (2024). Specifically, we sample 5% data from our training dataset to train the reference model. We follow the original implementation that focuses training on H→L tokens ( *i.e.,* the tokens with loss decreased from high to low during training the reference model) and masks the other tokens.

### B.2  IMPLEMENTATION DETAILS OF REWARDFT

For the RewardFT baseline, because of the lack of Agent preference data, we use general DPO data ORCA DPO (noa, 2024c) and Ultrafeedback (noa, 2024a) to train the model as token-level reward model under the same setting in (Rafailov et al., 2024). We calculate the token-level reward given by the preference model, then we follow the practice in weighted-MLE (Yang et al., 2023), taking softmax on all token rewards as the weight to train the model.

### B.3  IMPLEMENTATION DETAILS OF $\alpha$-FT

A simple and common method for addressing imbalance training is to manually give a fixed weight for each type of token (Lin et al., 2020). Here we introduce a weighting factor $\alpha \in [0, 0.5]$ for boilerplate tokens and $1 - \alpha$ for reasoning tokens. Let $\mathcal{L}_b$ and $\mathcal{L}_r$ represent the total loss for the boilerplate and reasoning tokens, respectively. The re-weighted loss (denoted as $\mathcal{L}_{\alpha-balance}$) can be formulated as follows:

$$\mathcal{L}_{\alpha-balance} = \alpha \mathcal{L}_b + (1 - \alpha) \mathcal{L}_r \qquad (7)$$

This loss is a simple extension to CE we call $\alpha$-FT in this paper that we consider as an experimental baseline for our proposed RFT method.

## C  MORE EXAMPLES LABELED BY SHAD

In Figure 8, we show several examples of tokens classified by SHAD method, with blue regions represent reasoning tokens and brown regions indicate boilerplate tokens.

## D  QUALITATIVE ANALYSIS

In this section, we present several examples of how model trained by our method yeild more accurate answer than model trained by naive SFT in Figure 9. In the response generated by the naïve SFT model, we observe overfitting, with formatted tokens (yellow) and template-connect tokens (blue) being erroneously generated. Additionally, the naïve SFT model exhibits hallucination, leading to reasoning errors (red). In contrast, our SHAD+RFT method successfully follows the Held-Out instructions and provides accurate reasoning.

## E  MORE EXAMPLE OF SHUFFLED DATA

In this section, we add more examples of shuffled data in Figure 10 to support the state made that shuffling the correspondence between input and output across data samples does not alter the predictability of boilerplate tokens while reasoning tokens are disruptive after the shuffling.

---

More Examples of Tokens Classified by SHAD

---

Thought: Based on the user's request for fresh manga content in the genres of Comedy, Romance, and Supernatural, I called the "fetch_latest_for_mangaverse_api" function from the MangaVerse API. With this function, I can retrieve the latest manga that matches the user's desired genres. By specifying the genres and requesting chapter lists and images, I will be able to assist the user in finding new manga to blog about. Now, I will call the "fetch_chapters_for_mangav erse_api" function to fetch the chapters for the manga that was fetched in the previous step. This will provide me with the necessary information to present to the user and further enhance their blogging experience.
Action: fetch_chapters_for_mangaverse_api
Action Input: {}

---

Thought: The current API, "businessplan_for_business_starter_api", is being called with the argument "{\n "idea": "Start a mobile pet grooming business"\n}". By calling this API with the user's selected business idea, I can generate a comprehensive business plan specifically tailored to the concept of starting a mobile pet grooming business. This will provide the user with a step-by-step guide on how to establish and run their venture, including details on services offered, target marke t, marketing strategies, and financial projections. The purpose of this API call is to assist the user in transforming their idea into a concrete and well-structured business plan.
Action: businessplan_for_business_starter_api
Action Input: {
  "idea": "Start a mobile pet grooming business"
}

---

{
 "tool_calls": [
   {"name": "live_players_rankings", "arguments": {"n_player": "300"}}, {"name": "teammedia", "arguments": {"is_id": 7890}}
  ]
}

---

{
 "tool_calls": [
   {"name": "products_search_by_barcode", "arguments": {"upc": "123456789012"}}, {"name": "get_product_s_reviews", "arguments": {"productid": "B012345678"}}
  ]
}

---

Figure 8: More case study of tokens classified by SHAD. The blue regions represent reasoning to-kens, identified by an increase in loss on the model tuned with shuffled data compared to the original model. In contrast, the brown regions indicate boilerplate tokens, characterized by a decrease in loss on the tuned model.

---

Examples on Held-Out Benchmark Nexus

---

**Instruction:**

You are given multiple functions and a user query.
Please proceed with generating a function call for the function with the proper arguments that best answers the given prompt.
**Respond with nothing but the function call ONLY,** such that I can directly execute your function call without any post processing necessary from my end. Do not use variables.
{tools}

─────────────── **Example 1** ───────────────

**User Question:**

Give me restaurants near me?

----------------------------------------------------------------------

**Response (SFT):**

Function Call: find_places_near_location(['restaurant'], 'my location', 10)
Explanation: Based on the user's query, I can see that they are looking for restaurants near their location. To fulfill this request, I will use the "find_places_near_location" function. The current arguments for the function call are "type_of_place" = ['restaurant'], "location" = 'my location'. These arguments indicate that the user is specifically interested in finding restaurants near their current location. The "radius_miles" argument is set to 10, which means that the search will be limited to a 10-mile radius. This will ensure that the recommendations are relevant to the user's current location.
```python
find_places_near_location(['restaurant'], 'my location', 10)
```

----------------------------------------------------------------------

**Response (SHAD+RFT):**

find_places_near_location(type_of_place=['restaurant'], location=get_current_location())

─────────────── **Example 2** ───────────────

**User Question:**

Give me vulnerabilities, sorted by their CVSS version 2.0 scores in descending order, released between January 1, 2021 and January 31, 2021, that match the CPE name 'cpe:/a:microsoft:windows_10:1909' and have a Technical Alert from US-CERT?

---

**Response (SFT):**

Function call: sortCVEsByCVSSv2Score(cve_list, descending=True)
Explanation: The user wants to search for vulnerabilities associated with the CPE named 'cpe:/o:linux:linux_kernel' and sort them in descending order by their CVSS Version 2.0 base scores. The function call 'sortCVEsByCVSSv2Score' is used to achieve this. The argument 'cve_list' is a list of CVE objects, and the argument 'descending' is set to True to sort the CVEs in descending order.

----------------------------------------------------------------------

**Response (SHAD+RFT):**

sortCVEsByCVSSv2Score(searchCVE(cpeName='cpe:/a:microsoft:windows_10:1909', pubStartDate='2021-01-01', pubEndDate='2021-01-31', hasCertAlerts=True), descending=True)

─────────────── **Example 3** ───────────────

**User Question:**

Could you help identify CPEs that closely resemble the pattern 'cpe:2.3:a:google:android:'? And please, limit the results to just 5.

---

**Response (SFT):**

Function call: searchCPE(cpeMatchString='cpe:2.3:a:google:android:', limit=5)
Explanation: The user wants to identify CPEs that closely resemble the pattern 'cpe:2.3:a:google:android:', and limit the results to just 5. To accomplish this, I will call the 'searchCPE' function with the arguments 'cpeMatchString' set to 'cpe:2.3:a:google:android:' and 'limit' set to 5. This will search for CPEs that match the given pattern and limit the results to just 5.

----------------------------------------------------------------------

**Response (SHAD+RFT):**

searchCPE(cpeMatchString='cpe:2.3:a:google:android:', limit=5)

---

Figure 9: **Comparison example on Held-Out Benchmark Nexus.** In the response generated by the naïve SFT model, we observe overfitting, with formatted tokens and template-connect tokens being erroneously generated. Additionally, the naïve SFT model exhibits hallucination, leading to reasoning errors. In contrast, our SHAD+RFT method successfully follows the Held-Out instructions and provides accurate reasoning, we explicitly mark the different reasoning part in red.

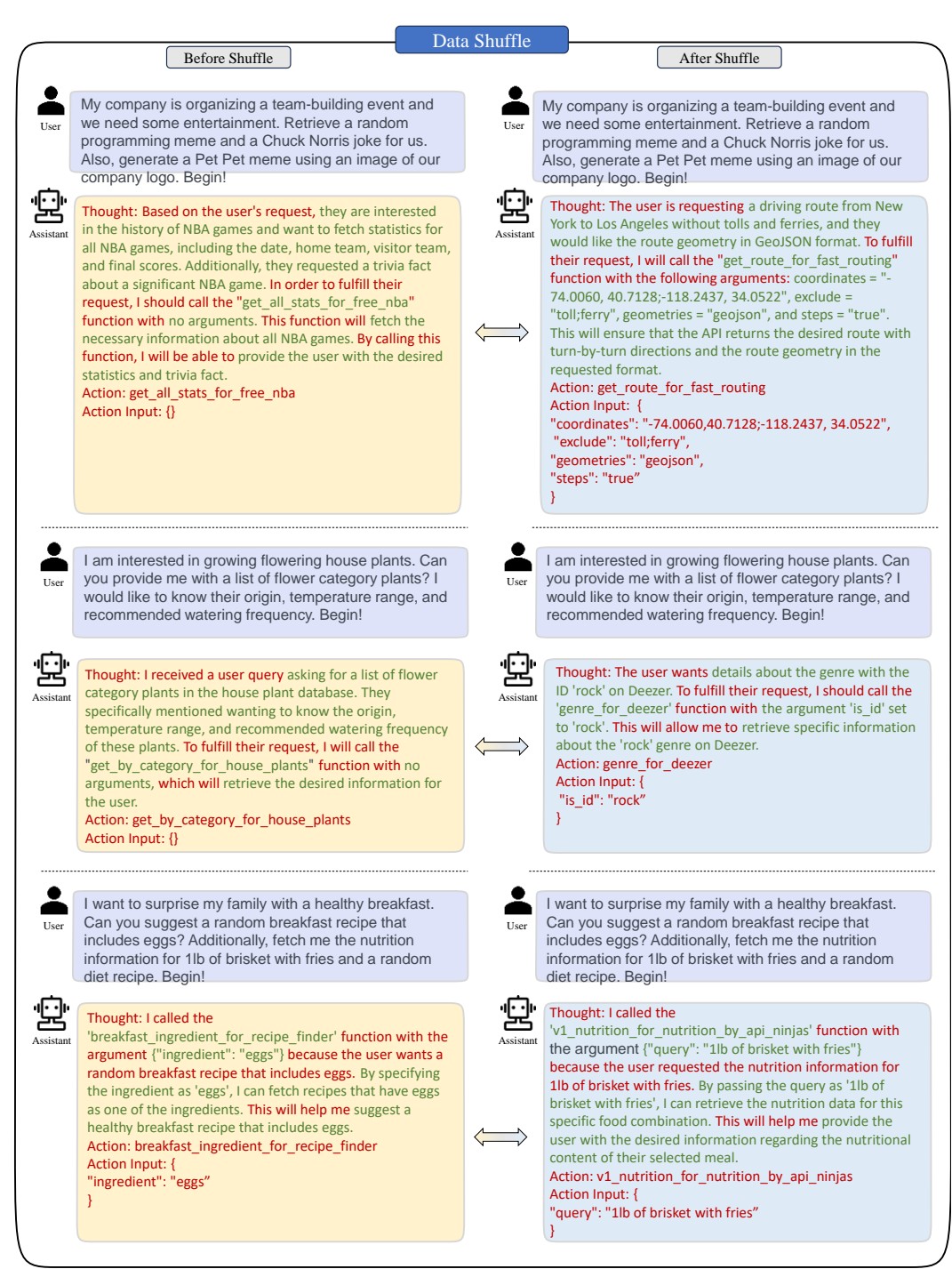

Figure 10: **More Example of Shuffled Data**. After shuffling, the assistant's responses no longer correspond to the original queries. However, some tokens (boilerplate tokens, red) remain semantically similar to the original response and are therefore predictable. In contrast, reasoning tokens (green) no longer align with the query, resulting in noise.

