# OpenReview forum: "Disentangling Reasoning Tokens and Boilerplate Tokens For Language Model Fine-tuning"
_ICLR.cc/2025/Conference — ICLR 2025 Conference Withdrawn Submission_

### Official Review · Reviewer_3z13 · 2024-10-25

**Soundness:** 2
**Presentation:** 2
**Contribution:** 2
**Rating:** 5
**Confidence:** 4

**Summary:**

This paper observes that reasoning tokens and boilerplate tokens (also known as format tokens) should be treated differently during fine-tuning; otherwise, models may easily overfit to boilerplate tokens. To address this issue, the authors propose a method called SHuffle-Aware Discriminator (SHAD), which is a data-driven approach designed to automatically identify boilerplate and reasoning tokens from the training set.

Building on the classification results obtained from SHAD, they further developed Reasoning-Highlighted Fine-Tuning (RFT), a variant of standard fine-tuning that assigns larger weights to reasoning tokens.

The proposed methods were tested on agentic benchmarks, particularly on function-calling benchmarks, to demonstrate their effectiveness.

**Strengths:**

1. The observation is interesting, but it haven't been verified in a larger scale, i.e., whether the differentiation of both tokens till mater when we scale up the diversity of the dataset.

2.  The authors compared with a very comprehensive list of baselines to demonstrate their effectiveness.

3. The idea of using probability changes to identify boilerplate tokens is simple, yet effective.

**Weaknesses:**

1. Related Work Discussion: Some related works should be discussed more thoroughly. One major contribution claimed by the authors is the intention to "emphasize the differences in learning difficulty and importance between reasoning and boilerplate tokens for agent learning." However, this concept has been discussed in previous research, such as Chen (2024b), but not adequately addressed in Section 2.1.

2. Effectiveness and Motivation Concerns: I have doubts regarding the effectiveness and motivation behind differentiating these tokens during training. As noted in the Limitations section, the effectiveness of the proposed approach relies on boilerplate tokens remaining consistent across different samples, as in the used training and evaluation datasets. This limitation could significantly impact the broader applicability of the proposed methods, as their effectiveness may diminish when scaling up the data. Could the authors conduct additional experiments to investigate this issue further?

3. From the right side of Figure 5, it appears that the proposed method sacrifices the learning of boilerplate tokens to enhance the learning of reasoning tokens. I am concerned that as we scale up the data—assuming we have sufficient diversity for both token types—this approach may ultimately harm overall performance.

**Questions:**

See weakness.

---

> ### Author Response · Authors · 2024-11-22
> **Response to Reviewer 3z13**
>
> Dear Reviewer 3z13,
>
> Thank you for dedicating your time and effort to reviewing our paper. We have carefully considered your feedback and address each concern below:
>
> **W1**. *Previous works, such as Agent-Flan (Chen, 2024b), have discussed the claimed contribution of emphasizing the differences between reasoning and boilerplate tokens for agent learning. The work has not been adequately addressed in the paper.*
>
> **A1**: Our contribution in discussing different types of tokens differs significantly from existing works, including the Agent-Flan paper. First, 1) The "boilerplate tokens" in existing works like Agent-Flan typically **refer only to formatting tokens and exclude the template-connecting tokens**, which are more challenging to separate from reasoning tokens. 2) More importantly, our contribution **goes beyond highlighting these differences**—it emphasizes the necessity of distinguishing (classifying) these tokens, a point not discussed in existing works like Agent-Flan. This marks a fundamental difference in the problems tackled and the solutions proposed. We focus on automatically disentangling reasoning tokens from boilerplate tokens, Agent-Flan prioritizes converting agent data into a standard conversational format. We will update our contribution statement and related work discussion to clarify these distinctions more explicitly.
>
> **W2**: *The effectiveness of the proposed approach relies on boilerplate tokens remaining consistent across different samples. This limitation could significantly impact the broader applicability of the proposed methods, as their effectiveness may diminish when scaling up the data.*
>
> **A2**: We acknowledge that our method's effectiveness may decrease as the diversity of boilerplate tokens increases. **However, in agent data collection, achieving the high diversity is inherently challenging. **This is because such data collections typically adhere to specific formats/templates (e.g., Agent-FLAN, AgentTuning, ToolLLaMA, FireAct, and ApiGen), as deviating from these standards significantly increases collection costs and complicates quality control. **Simply scaling up the data size does not significantly increase the diversity of boilerplate tokens** due to the inherent constraints of collection paradigms. **Moreover, even when diversity is enhanced** through specific approaches, such as mixing datasets with different formats/templates (as demonstrated in this study with ToolLLaMA and ApiGen), our method continues to perform well because our shuffle-based processing can be applied within data sharing similar formats/templates.
>
>
> **W3**: *From the right side of Figure 5, it appears that the proposed method sacrifices the learning of boilerplate tokens to enhance the learning of reasoning tokens. As scaling up the data—assuming having sufficient diversity for both token types—this approach may ultimately harm overall performance.*
>
> **A3**: In the figure, the sacrifice in learning boilerplate tokens is very small compared to the improvement in learning reasoning tokens, **so overall, our approach can enhance the learning**. Additionally, boilerplate tokens are relatively easier to learn, meaning the impact of slight sacrifices in learning is likely to be smaller, even no negative impact, as such a sacrifice may actually indicate a reduction in overfitting to the token.
> Regarding scaling up to increase the diversity of both token types, first, as discussed in A2, increasing the diversity of boilerplate tokens is inherently difficult; Second, even if diversity is increased, our method will adaptively adjust the weight (see Equations (5) and (6)) for the identified token types based on their loss, thereby reducing the challenges associated with the learning sacrifice.

---

> > ### Comment · Reviewer_3z13 · 2024-11-25
> >
> > I understand the nuanced differences in the definition of 'boilerplate tokens' between previous works and our paper. Although these papers lead to different solutions and ultimately focus on distinct aspects, they share remarkably similar motivations and thus warrant thorough discussion in the related work section.
> >
> > Regarding W2's concern, while the listed datasets indeed have limited diversity, this does not necessarily mitigate the potential methodological limitations. Many high-quality, manually collected agentic datasets exist with significant diversity. Moreover, contemporary research in model-based data generation increasingly emphasizes data diversity. To address this concern comprehensively, I recommend incorporating statistical evidence—for instance, demonstrating that even in high-quality human-collected datasets, a substantial proportion of boilerplate tokens persist, and/or showing that the model maintains robust performance despite these characteristics.

---

> > > ### Author Response · Authors · 2024-11-25
> > >
> > > Thank you for your reply. We have included the relevant detailed discussion in the related work section, and the revised version will be uploaded later. Regarding W2, **could you please provide the name of the manually collected agentic dataset with significant diversity?**

---

> > > > ### Author Response · Authors · 2024-12-01
> > > > **Updated related work discussion regarding Agent-flan**
> > > >
> > > > Dear Reviewer,
> > > >
> > > > We have uploaded a revised version of the paper (28 Nov), which now includes an expanded discussion of the Agent-flan paper in the Related Work section. The detailed changes are as follows:
> > > >
> > > > ```
> > > > Among existing works, Agent-Flan (Chen et al., 2024b) is the most relevant to ours, sharing a similar motivation to account for token differences in agent tuning. However, it only considers “format tokens” as boilerplate tokens, overlooking template connecting tokens, which are more challenging to disentangle from reasoning tokens. Additionally, it does not emphasize the importance of distinguishing (or classifying) these tokens, resulting in a fundamental difference in both the problems addressed and the solutions proposed. We focus on automatically disentangling reasoning tokens from boilerplate tokens, whereas Agent-Flan prioritizes converting agent data into a standard conversational format.
> > > > ```

---

> > > ### Author Response · Authors · 2024-11-26
> > > **Inquiry on Agentic Datasets with Significant Diversity**
> > >
> > > We agree that "high-quality, manually collected agentic datasets with significant diversity" can enhance our method's evaluation. However, we have not been able to locate such datasets. Could you kindly share information about datasets that meet these criteria?

---

> > > > ### Comment · Reviewer_3z13 · 2024-11-27
> > > >
> > > > There are many such datasets out there, e.g., OSWorld and AndroidControl. But I am not proposing that authors should conduct exp on them, since that would need substantial methodological adaptations to work with those benchmarks. Just want to hear your thoughts on the effectiveness of applying the proposed method to such datasets, which will help clarify the broader applicability of this work.

---

> > > > > ### Author Response · Authors · 2024-12-01
> > > > > **Response to the OSWorld and AndroidControl Datasets**
> > > > >
> > > > > Dear Reviewer,
> > > > >
> > > > > We sincerely appreciate you highlighting the OSWorld and AndroidControl datasets. These datasets represent high-quality, manually collected multimodal agent data. The tasks involved are set in complex environments and often demand robust multi-step execution capabilities, making these datasets valuable benchmarks for agentic systems.
> > > > >
> > > > > After reviewing these datasets, we believe our method can still effectively distinguish text tokens within them. **Specifically, the core assumption underlying the effectiveness of our method is the presence of a certain level of uniformity in agent framework prompts and structured outputs, which holds true for these datasets (to some degree) due to their highly standardized interaction formats.** Below, we explain how the characteristics of these datasets align with our assumptions, ensuring the continued applicability of our method:
> > > > >
> > > > > 1. **OSWorld**: This dataset employs fixed prompt templates (e.g., *"PROMPT for Common Setting"* and *"PROMPT for SoM Setting"*) to query the model. The outputs follow predefined rules, often resulting in default (frequent) operations such as inserting *"time.sleep(0.5)"* between actions. Furthermore, the outputs are converted into structured JSON formats, introducing formatting-related tokens. These default operations and formatting tokens demonstrate high consistency across samples, reducing their learning difficulty and functioning as "boilerplate" elements. Our method remains effective in distinguishing these tokens from reasoning-related tokens.
> > > > >
> > > > > 2. **AndroidControl**: Prompts like *"SeeAct"* and *"ER"* are designed to standardize task descriptions. Then, the model is required to generate structured JSON data in formats such as `{"action_type":"click","x":<x_coordinate>,"y":<y_coordinate>}`. This closely resembles the structured outputs in our study (e.g., *ApiGen*), where the formatting of actions is consistent across examples. Our method can effectively identify these consistent components.
> > > > >
> > > > > In conclusion, while these datasets involve increased complexity, their standardized interaction protocols ensure that certain (text) boilerplate elements are still shared across samples. Our method remains effective in identifying these boilerplate **text tokens**. However, we acknowledge that, for tokens that become more diverse yet are not reasoning-related, the method's discrimination capability does decrease. As mentioned in the *"Limitations"* section, identifying similar samples for such tokens and applying our method to these subsets may help maintain its effectiveness, offering a promising direction for future improvements.
> > > > >
> > > > > Regards,
> > > > >
> > > > > Authors

---

> > > > > > ### Comment · Reviewer_3z13 · 2024-12-01
> > > > > >
> > > > > > Thanks for sharing your thoughts and adding related work discussion, I will increase my score.

---

### Official Review · Reviewer_896s · 2024-10-30

**Soundness:** 2
**Presentation:** 3
**Contribution:** 2
**Rating:** 3
**Confidence:** 4

**Summary:**

This paper proposes to enhance LLM fine-tuning by disentangling the reasoning and boilerplate tokens, implemented by discriminating the two types of tokens with loss differences first, then adaptively optimizing the differential weights of different tokens. Experimental results on several tool-use datasets show that the proposed strategy can outperform naïve SFT.

**Strengths:**

1. Introducing token-wise loss to LLM agents and fine-tuning with differential weights are interesting, and empirical results on some datasets demonstrate the initial premise.
2. The paper's writing is well and easy to understand.

**Weaknesses:**

1. Apart from the concrete definition of different token types, there is no ground truth data for classifying the reasoning and boilerplate tokens. It would be convincible if the authors could showcase the accuracy of the token classifier, displaying a few cases that are not that universal.
2. As the author presented in lines 74-76 shuffling can cause reasoning tokens mismatching, however, there are no proofs for such a statement.
3. There are only 8B models for empirical evaluation, the proposed approach should be validated on larger models to demonstrate the generality.
4. Not enough explanation for different experimental results and ablation studies.
5. There are typos, such as “Classifiying” in line 211, and the order repeated error in the Compared paragraph.

**Questions:**

1. Why agent capabilities (i.e., multi-step reasoning and tool-use) are relevant to the reasoning tokens? Sometimes, the instructions and their corresponding tools can be viewed as a kind of commonsense knowledge rather than reasoning.
2. In Figure 1, the authors colored the reasoning and boilerplate tokens for a given instance, however, it confused me. How to choose boilerplate and reasoning tokens for shuffling? And why “find the most popular genre”, “analyze the”, “currently”, and “the most” tokens are reasoning tokens? These tokens are also crucial for reasoning and can be considered as reasoning tokens.
3. Are these training losses in Figure 5-right from LLaMA-3-8B? Such a phenomenon cannot prove the adaptivity of other models.
4. How many conversation data were sampled from ShareGPT and what is the ratio of general conversation and reasoning data? As ShareGPT has already included a lot of reasoning data, it may affect the SFT performance.
5. It seems that RFT loss and SFT loss eventually overlapped after 2000 training steps, and SFT loss decreases faster, but the evaluation accuracies of these methods are way from each other, why did that happen?

---

> ### Author Response · Authors · 2024-11-22
> **Response to Reviewer 896s: Part 1(weaknesses)**
>
> Dear Reviewer 896s,
>
> Thank you for dedicating your time and effort to reviewing our paper. We have carefully considered your feedback and address each concern below:
>
> **W1**: *Apart from the concrete definition of different token types, there is no ground truth data for classifying the reasoning and boilerplate tokens. It would be convincible if the authors could showcase the accuracy of the token classifier, displaying a few cases that are not that universal.*
>
> **A1**: We acknowledge the lack of ground-truth labels for classifying the two types of tokens, as no open-access dataset provides such labels. Manual annotation is also impractical due to the labor-intensive process and the inherent difficulty of distinguishing between token types—boilerplate tokens often intertwined with reasoning tokens, making manual labeling infeasible. As a result, we cannot obtain such labels or use them to compute classification accuracy. However, we did attempt to label the portion of boilerplate tokens that can be identified using regular expressions (i.e., the format part). Our method achieves a very low error rate in distinguishing this portion (less than 1%).
> Moreover, to evaluate the effectiveness of token classification, we have conducted an indirect evaluation through the performance of a downstream fine-tuning task (using the classified results to enhance the learning weight of reasoning tokens). With our classification, we achieve improved performance, which also validates the effectiveness of our classification method.
>
> **W2**:  *As the author presented in lines 74-76 shuffling can cause reasoning tokens mismatching, however, there are no proofs for such a statement.*
>
> **A2**: It seems that this issue arises from a potential misunderstanding of the shuffle process. We apologize for any confusion caused.
> Our shuffling operation is performed by directly permuting question-response pairs. For example, assuming there are three training samples (Q_1, R_1), (Q_2, R_2), (Q_3, R_3), one shuffle result could be (Q_1, R_3), (Q_2, R_1), (Q_3, R_2), where Q and R represent the question and response, respectively. After shuffling, the reasoning in response (e.g., R_3) no longer addresses the question (e.g., Q_1). Figure 3 in the paper provides a detailed example, showing how reasoning components (e.g., function names and parameters) are mismatched with the corresponding questions after shuffling, while non-reasoning parts (e.g., connecting words) retain a high degree of similarity.
>
> **W3**: *There are only 8B models for empirical evaluation, the proposed approach should be validated on larger models to demonstrate the generality.*
>
> **A3**: Thanks for the suggestion. Due to resource constraints, we have focused our experiments primarily on 8B models. Given that models in the 7–8B size range are a common choice for real-world applications, we believe the current evaluation could demonstrate the value of our method. However, we agree that incorporating larger-scale validations would further strengthen this work, and we plan to incorporate such validations when sufficient resources become available.
>
> **W4**:  *Not enough explanation for different experimental results and ablation studies.*
>
> **A4**: Thank you for pointing this out. In the submitted version, we focused on presenting only the essential conclusions and explanations to keep the paper concise. In the revised version, we will expand on the explanations of the experimental results and ablation studies, e.g., including a more detailed discussion comparing our method with difference baselines.
>
> **W5**: *There are typos, such as “Classifiying” in line 211, and the order repeated error in the Compared paragraph.*
>
> **A5**: Thank you for pointing out these issues. We will carefully review the manuscript and fix the typos and the repeated errors.

---

> > ### Comment · Reviewer_896s · 2024-11-24
> > **Re-Response to the weaknesses**
> >
> > Thanks for the authors' responses, I have read them carefully with the manuscript. Unfortunately, none of my concerns have been addressed except for the typos.
> >
> > For the reasoning and boilerplate tokens, if annotating for all data is impractical, labeling a small proportion can not cost much, and it can also display the classification performance; or there should be more case studies to illustrate the consistency.
> >
> > Weakness 2 was caused by the joint definition and classification of the two token types, rather than the shuffle operation.
> >
> > Mere 7-8B models cannot demonstrate the generalization of a method, various scales can enhance the faithfulness.
> >
> > There is no detailed introduction for different experimental results and ablation studies yet.

---

> ### Author Response · Authors · 2024-11-22
> **Response to Reviewer 896s: Part 2(question)**
>
> **Q1**: *Why agent capabilities (i.e., multi-step reasoning and tool-use) are relevant to the reasoning tokens? Sometimes, the instructions and their corresponding tools can be viewed as a kind of commonsense knowledge rather than reasoning.*
>
> **A1**: To our knowledge, existing works typically consider multi-step reasoning and tool use as integral to reasoning abilities and essential components for reasoning evaluation. In general, multi-step reasoning directly reflects the reasoning process of breaking down complex problems into logical, sequential steps. As for tool use, it extends beyond common knowledge and is also tied to reasoning. For example, some tools may not be part of common knowledge and can only be utilized by understanding their descriptions. Leveraging these tools requires reasoning abilities to effectively match them with specific problems.
>
> **Q2**: *In Figure 1, the authors colored the reasoning and boilerplate tokens for a given instance, however, it confused me. How to choose boilerplate and reasoning tokens for shuffling? And why “find the most popular genre”, “analyze the”, “currently”, and “the most” tokens are reasoning tokens? These tokens are also crucial for reasoning and can be considered as reasoning tokens.*
>
> **A2**: Sorry for the misunderstanding. Our shuffling approach does not require selecting boilerplate or reasoning tokens. As shown in Figure 2, shuffling is performed directly by permuting question-response pairs. For instance, given three samples \((Q_1, R_1), (Q_2, R_2), (Q_3, R_3)\), one possible shuffle result is \((Q_1, R_3), (Q_2, R_1), (Q_3, R_2)\), where \(Q\) and \(R\) denote the question and response, respectively. Thus, selecting boilerplate and reasoning tokens is not needed for our shuffling process.
>
> **Q3**: *Are these training losses in Figure 5-right from LLaMA-3-8B? Such a phenomenon cannot prove the adaptivity of other models.*
>
> **A3**: Yes, Figure 5 (right) shows the training loss for LLaMA-3-8B. Regarding the training loss of other models, the training logs do not include loss calculations for different tokens, so we need to rerun the experiments. Due to limited computational resources recently, we have not been able to allocate resources for this experiment. We will do our best to include the corresponding results before the rebuttal DDL.
>
> **Q4**: *How many conversation data were sampled from ShareGPT and what is the ratio of general conversation and reasoning data? As ShareGPT has already included a lot of reasoning data, it may affect the SFT performance.*
>
> **A4**: Following previous works, all data from ShareGPT is utilized, with the ratio of ShareGPT to agent-specific data being approximately 3:1. Although ShareGPT includes some reasoning-related questions, these are purely general question-answer pairs and lack the characteristics of agent-specific data. Furthermore, all compared methods use the same ShareGPT data, ensuring no differences in reasoning abilities derived from it.
>
> **Q5**: *It seems that RFT loss and SFT loss eventually overlapped after 2000 training steps, and SFT loss decreases faster, but the evaluation accuracies of these methods are way from each other, why did that happen?*
>
> **A5**: Although the overall loss in the convergence state appears similar (with RFT being slightly lower), RFT's adaptive re-weighting consistently prioritizes reasoning tokens, strengthening their learning. This approach further reduces the loss of reasoning tokens, though it slightly compromises the performance on other tokens that are already well-fitted (See Figure 5 Right). The enhanced learning of reasoning tokens improves task performance.

---

> ### Author Response · Authors · 2024-11-24
> **Clarifications on Our Response to Weaknesses**
>
> Dear Reviewer,
>
> We regret that our previous response did not meet your expectations. However, we would like to provide clarifications for the concerns raised:
>
> 1. **Regarding Weakness 1:**
>      - As stated in our initial response, we have annotated the easily identifiable boilerplate tokens (formatting tokens) in our approach, **achieving a misclassification rate of less than 1%**.
>       - Conducting token-by-token annotations for samples is challenging due to subjectivity and the high cost of ensuring quality even in small-scale annotations. We believe providing more case studies is a more practical and robust alternative. **We have additional case studies that can demonstrate consistency, which will be included in the revised paper to be uploaded as soon as possible.**
>
>
> 2. **Regarding Weakness 2:** Our misunderstanding of your comments stemmed from Question 2, and we sincerely apologize for the oversight. We would like to restate our response here — the mismatch of reasoning tokens after "shuffling" is evident. While it is difficult for us to provide theoretical proof, there is **substantial empirical evidence** supporting this claim:
>      - Agent datasets are typically generated using structured response formats specific to their frameworks (e.g., Agent-FLAN, AgentTuning, ToollLaMA, FireAct, and ApiGen). This approach results in **highly consistent formats and reasoning styles (template-connecting tokens)** across tasks within the dataset, as demonstrated in **existing studies** (e.g., Agent-FLAN).
>     - **Practical examples**, such as those shown in Figure 3 of our paper, demonstrate that shuffling causes the reasoning parts of the response to become mismatched with the questions, e.g., ```function names/parameters — correct one: ultimateoscillator_for_qvantana*, after shuffling: *maxindex_for_twelve_data)```,
> while boilerplate tokens remain highly similar, e.g., ``` connecting worlds --- *before shuffling: "Thought: I need to call... This API call will... this will help me...", after: "Thought: I should call... This API call is... this will help me..."```.
>
> 3. **Regarding larger LLM models:** We currently do lack sufficient resources to validate larger models. However, we will make every effort to conduct validation on models of other sizes (specifically at the 3B level) during the rebuttal period.
>
> 4. **Regarding analyses on experimental results:** We have incorporated specific updates in the revised paper. However, we have not yet uploaded it as we are still completing the experiments suggested by the reviewers. We will upload the revised version as soon as possible. These updates include a more detailed analysis of baseline comparisons and discussions regarding the limited performance of ablation studies.
>
> We hope that our updated feedback addresses your concerns. Please let us know if you have any further questions or concerns.
>
> Sincerely,
> Authors

---

> ### Author Response · Authors · 2024-12-01
> **Follow up and updtaes**
>
> Dear Reviewer,
>
> We would like to follow up on our previous clarifications and inquire if you have any feedback on them. Below are the updates regarding the case study and experiment points mentioned in the clarifications:
>
> We have uploaded a revised version of the paper, which includes updates to address the weaknesses you raised:
>
> 1) To address Weakness 1, we have added more case studies on the classification results and model outputs. Besides the study on the classification results, the case studies on model outputs are further used to illustrate the effectiveness of highlighting the learning of reasoning tokens identified by our method. The results show that by emphasizing the learning of reasoning components identified by our classification method, we can consistently enhance the model's ability to correctly apply functions (the reasoning part), such as providing accurate parameters, while avoiding overfitting the training format. All case studies can be found in the experimental section and appendix.
>
>
> 2) For Weakness 2, when first introducing the "shuffling can cause reasoning tokens mismatching" claim, we have included a footnote (Footnote 1) that links to the evidence supporting the claim, found in the method section and appendix.
>
> 3) Regarding the analyses of experimental results, we have expanded the discussion, and these additions are highlighted in blue.
>
> Regarding the experiments with the 3B model, we have obtained computational resources and are in the process of generating results. We will upload the results as soon as they are available.
>
> Thank you for your time and consideration.
>
> Best regards,
> The Authors

---

> > ### Comment · Reviewer_896s · 2024-12-01
> > **Re-Follow-up and updates**
> >
> > Thanks for the authors' follow-up, I have read both the latest responses and the manuscript.
> >
> > Albeit the 1% misclassification rate was not supported by experimental results, the authors did provide more case studies to illustrate their premise more clearly, also, more explanations have been complemented in the experimental analysis part.
> >
> > However, there are still no more results from the proposed method with LLMs of different architectures and scales, leaving concerns for generality and scalability. Furthermore, as both the proposed method and baselines are fine-tuning derivatives, experiments with more hyper-parameters would be better to demonstrate the robustness.

---

### Official Review · Reviewer_PEUP · 2024-11-04

**Soundness:** 3
**Presentation:** 3
**Contribution:** 3
**Rating:** 6
**Confidence:** 3

**Summary:**

This paper addresses the challenge of fine-tuning Large Language Models (LLMs) for agent capabilities by introducing a novel approach to token differentiation during training. The authors observe that in agent-task datasets, tokens serve different roles - specifically reasoning tokens (which contain task-specific logic) and boilerplate tokens (which handle output formatting and standard transitions). The authors' empirical analysis shows that their method effectively identifies reasoning tokens and enhances their learning while maintaining performance on boilerplate tokens, leading to improved overall agent capabilities in LLMs while preserving generalization ability.

**Strengths:**

This paper demonstrates notable strengths across several dimensions. In terms of originality, it introduces a fresh perspective on token differentiation in agent training through its novel SHAD method and adaptive weighting mechanism, being the first to explicitly address the distinction between reasoning and boilerplate tokens. The quality of the work is evident in its comprehensive evaluation across multiple benchmarks, well-designed ablation studies, and clear empirical evidence supported by thorough loss analysis and effective visualizations. The clarity of the paper is commendable, featuring a well-structured presentation, clear explanations supported by helpful diagrams, accessible mathematical formulations, and illuminating examples and case studies. The significance of the work is substantial, as it addresses a fundamental challenge in agent training, demonstrates meaningful improvements over existing methods, offers a practical and implementable approach, and provides insights that could extend beyond agent training. What makes this paper particularly strong is how it combines novel conceptual insights with practical implementation, all while maintaining clarity and reproducibility in its presentation.

**Weaknesses:**

While the shuffle-based discrimination method is innovative, it lacks theoretical foundations and formal guarantees, with arbitrary thresholds for token classification. There's also a noticeable absence of qualitative analysis demonstrating how improved reasoning manifests in actual outputs.

**Questions:**

Could you provide a qualitative analysis demonstrating how improved reasoning manifests in actual outputs?

---

> ### Author Response · Authors · 2024-11-25
> **Response to Reviewer PEUP**
>
> Dear Reviewer,
>
> Thank you for your positive feedback on our approach and contributions. We appreciate your thoughtful comments and respond to your concerns as follows:
>
> ---
>
> ### **W1: Theoretical Foundations and Threshold Selection for Token Classification**
>
> **A1:** Thank you for raising these points. We agree that developing a rigorous theoretical framework would significantly strengthen this work. However, given the complexity of the problem and the early stage of exploration, achieving this goal remains challenging. In this study, we primarily focus on pointing out the problem and presenting a solution informed by empirical observations, leaving the development of theoretical foundations for future work.
>
> Regarding threshold selection, while it may appear somewhat arbitrary, it is empirically justified. Boilerplate and reasoning tokens exhibit distinct predictability behaviors when input-output combinations are shuffled across samples. Specifically, boilerplate tokens retain predictability, whereas reasoning tokens do not. This observed behavior underpins our approach, where tokens are classified based on whether their loss change (after shuffling) exceeds zero.
>
>
>
> ### **W2: Qualitative Analysis of Improved Reasoning Outputs**
>
> **A2:** Thank you for this valuable suggestion. We have added case studies to demonstrate how our methods improve reasoning in the function-call datasets. In general, our approach enhances the model's ability to correctly use functions (reasoning part), such as by providing accurate parameters, and avoids overfitting to the format observed in training data. Below are examples:
>
> **Example 1:**
>
> ```plaintext
> ## Instruction: You are given multiple functions and a user query. Please proceed with generating a function call for the function with the proper arguments that best answer the given prompt. Respond with nothing but the function call ONLY, such that I can directly execute your function call without any post-processing necessary from my end. Do not use variables.
> {tools}
> Question: Give me restaurants near me.
>
> ## Response (SFT):
> Function Call: find_places_near_location(['restaurant'], 'my location', 10)
> Explanation: Based on the user's query, I can see that they are looking for restaurants near their location. To fulfill this request, I will use the "find_places_near_location" function. ...
>
> ## Response (Ours):
> find_places_near_location(type_of_place=['restaurant'], location=get_current_location())
> ```
>
> **Analysis:** The user's query asks for restaurants nearby. With standard SFT, the model hallucinates a user location ("my location, 10") instead of invoking the correct function (get_current_location()) to retrieve it, as our method does. Accurate function usage directly relates to reasoning, demonstrating our method's ability to improve reasoning outputs.
>
> Additionally, the instruction explicitly requires --- *"Respond with nothing but the function call ONLY."* The model obtained by SFT tends to include explanations alongside the function call, following the format seen in training. In contrast, our method adheres strictly to the instruction, avoiding overfitting to training formats.
>
> ---
>
> Additional examples will be included in the Appendix of the revised paper, and we will upload the revision as soon as possible.

---

> ### Author Response · Authors · 2024-11-29
> **Updated version with case studies on model outputs**
>
> Dear Reviewer,
>
> We have submitted a revised paper. In Section 5, we have added a case study (more results can be found in the Appendix)  comparing the model outputs of naive SFT and our method. The results demonstrate that our approach improves the correctness of reasoning components (e.g., accurate function calls) while mitigating overfitting to the training format.
>
> Best regards,
> The Authors

---

> ### Author Response · Authors · 2024-12-01
> **Could you let us know if our rebuttal has sufficiently addressed your concerns?**
>
> Dear Reviewer,
>
> As the discussion period is nearing its end, we wanted to follow up and check if our responses have fully addressed your concerns.
>
> Thanks,
>
> Authors

---

### Note · Authors · 2024-12-09

**Comment:**

We sincerely thank the reviewers for their valuable feedback on our paper. Considering the low final scores and the need to include additional experimental results, we have decided to withdraw the submission.

**Withdrawal Confirmation:**

I have read and agree with the venue's withdrawal policy on behalf of myself and my co-authors.